# Women during Lactation Reduce Their Physical Activity and Sleep Duration Compared to Pregnancy

**DOI:** 10.3390/ijerph191811199

**Published:** 2022-09-06

**Authors:** Gema Cabrera-Domínguez, María de la Calle, Gloria Herranz Carrillo, Santiago Ruvira, Pilar Rodríguez-Rodríguez, Silvia M. Arribas, David Ramiro-Cortijo

**Affiliations:** 1Department of Physiology, Faculty of Medicine, Universidad Autónoma de Madrid, C/Arzobispo Morcillo 2, 28029 Madrid, Spain; 2Department of Obstetric and Gynecology, Hospital Universitario La Paz, Paseo de la Castellana 261, 28046 Madrid, Spain; 3Division of Neonatology, Hospital Clínico San Carlos, Instituto de Investigación Sanitaria del Hospital Clínico San Carlos (IdISSC), C/Profesor Martin Lagos s/n, 28040 Madrid, Spain; 4Food, Oxidative Stress and Cardiovascular Health (FOSCH) Research Group, Universidad Autónoma de Madrid, Ciudad Universitaria de Cantoblanco, 28049 Madrid, Spain; 5PhD Programme in Pharmacology and Physiology, Universidad Autónoma de Madrid, 28049 Madrid, Spain

**Keywords:** physical activity, pregnancy, lactation, quality of life, sleep, psychological capital

## Abstract

Sleep, mental health and physical activity are fundamental for wellbeing, and some of these factors are interrelated. However, these aspects are not usually considered during pregnancy and lactation, which are particularly vulnerable periods. Therefore, our aims were to conduct a cross sectional study to assess the psychological capital, quality of life, sleep hygiene and physical activity in a cohort of women during pregnancy and lactation periods. Women were recruited from Spanish maternity and lactation non-profit associations and social networks through an online platform with the following inclusion criteria: pregnancy (in any period of gestation) or breastfeeding period (≤6 months postpartum). The cohort was categorized into ≤12 weeks of gestation (n = 32), >12 weeks of gestation (n = 119) and lactation (n = 60). The women self-reported the sociodemographic data, obstetric complications and full breastfeeding or mixed practices. In addition, women responded to the psychological capital instrument, the health survey form, the Pittsburg sleep quality index and the pregnancy physical activity questionnaire. Overall, the groups were similar in sociodemographic variables. Women in the lactation period perceived lower social support compared to the gestation period. No statistically differences were found between groups in the psychological capital nor in the general health survey form. However, the models adjusted by employment and civil and economic status and perceived social support, demonstrated that the sleep duration negatively associated with the lactation period (β = 1.13 ± 0.56; *p*-Value = 0.016), and the household tasks were associated with this period (β = 2147.3 ± 480.7; *p*-Value < 0.001). A decrease in physical daily activities were associated with both the end of gestation and the lactation periods. In addition, the decreasing total activity was associated with the lactation period (β = 1683.67 ± 688.05; *p*-Value = 0.016). In conclusion, during lactation, the poorer sleep and physical activity, together with a lower social support of the woman, may lead to deficient mental health adjustment. Our data suggest that women are at higher risk of vulnerability in lactation compared to the gestation period.

## 1. Introduction

The maternity is an important period of life, not only due to the biological changes, but also the social impact. Therefore, pregnancy and lactation are singular periods where women’s health must be watched over from an integral point of view. Factors, such as worries, stress, anxiety and discomfort exert an impact on the quality of women’s life during pregnancy [1] and lactation [2] and, therefore, on maternal-neonatal health [3]. 

During the lactation period, health care is directed towards the newborn, reducing attention to the mother, who is in a critical period shifting to a new role in society. During this period, aspects, such as mental and physical health, sleep quality or physical activities could be forgotten. These factors are also key for the well-being of woman and the newborn.

The psychological adjustment of the woman during pregnancy and lactation is closely related to well-being, cognitive appraisals and coping with stress life events [4]. Thus, it has been shown that maternal self-efficacy is a predictor for a better psychological adjustment after miscarriage [5]. Additionally, there are positive associations between psychological parameter and maternal and neonatal health. For example, hope is positively associated with maternal attachment to the fetus [6], optimism, breastfeeding adherence and resilience with birth weight [7,8]. Self-efficacy, hope, optimism and resilience belong to the components of psychological capital, a construct that derives from positive psychology [9]. However, psychological capital has been scarcely used in health psychology in the field of maternity.

Sleep is another key aspect, markedly influencing life quality. During pregnancy and lactation, deficient sleep duration and quality are commonly reported [10], which diminishes the woman well-being [11]. Additionally, during the lactation period, women often complain of fatigue and shortened sleeping time due to childcare [12,13]. Nearly 64% of mothers are affected by fatigue during postpartum period, being a common issue during the period of adaptation to maternity. Postpartum fatigue can negatively affect the mother-infant interaction and, thus neonatal health and postpartum maternal recovery [14]. 

Together with psychological adjustment and sleep hygiene, physical activity is another important aspect for women during pregnancy and in the postpartum period. Additionally, it may also influence in the mentioned factors. For example, during lactation period, physical activity can improve sleep quality increasing vitality and reducing postpartum maternal fatigue [15]. A link between regular physical activity and psychological well-being, has also been reported, improving self-efficacy and quality of life and reducing anxiety and depression [16]. 

Additionally, light-to-moderate exercise in pregnancy is not associated with risks for the newborn; instead, it can lead to changes in lifestyle that imply long-term benefits [17,18]. Physical activity usually decreases during the postpartum period [16], likely related to the larger proportion of care-giving tasks for the mother [19]. Therefore, given the benefits of physical exercise during this period of life, more studies are needed to evaluate the impact of physical activity, particularly in lactation [20], for correct counselling.

With the abovementioned background, our aim was to study the differences in (1) psychological capital, (2) quality of life, (3) sleep hygiene and (4) physical activity among women at the beginning of pregnancy, at the end of pregnancy and during lactation period. This will allow increasing knowledge in areas of health during maternity that have been little explored.

## 2. Materials and Methods

### 2.1. Study Design and Women Enrollment

This observational and cross-sectional study investigated women during single pregnancy and lactation, using an online questionnaire comprising five sections: (1) sociodemographic questions, (2) psychological capital tool (PsyCap), (3) pregnancy physical activity questionnaire (PPAQ), (4) health survey form (SF-36) and (5) the Pittsburgh sleep quality index (PSQI). In addition, women during lactation completed a questionnaire regarding breastfeeding practices to obtain a breastfeeding adherence score (BAS). 

This study was conducted according to the principles of the Declaration of Helsinki, with the approval of the Ethical Committee of Universidad Autónoma de Madrid (Ref. CEI-124-2543). All online data were anonymously collected and no IP addresses were recorded, the participation was voluntary enrolled between November 2021 to June 2022, and women had the possibility to end their survey at any time without saving any of their responses.

The sample size was estimated in 125 women, considering a F-distribution under null hypothesis, an effect size of 0.3, a probability alpha of 5% and a statistical power of 85%. For this calculation was used the software G*Power (version 3.1.9.6, Franz Faul, Kiel University, Kiel, Germany). After considering a 15% of missing data, the adjusted sample size was to enroll, at least, 147 women.

Women were recruited through non-profit breastfeeding associations, social networks and maternity-specific discussion boards. The inclusion criteria were internet access, willingness to participate in the study, Spanish language comprehension and to be single pregnant or being in the first 6 months postpartum (lactation period). Exclusion criteria were lack of answer or typographical errors in the database. 

Furthermore, to avoid biases with previous morbidities that could interfere with psychological capital, the practice of physical activity or sleep, women with hypertension, high cholesterol and/or triglycerides, obesity, diabetes mellitus, asthma, cancer, autoimmune diseases and depression under pharmacological treatment were not included in the study. Mothers of twin infants meeting these criteria were asked to complete only one questionnaire to avoid duplication. The questionnaire was administered in Spanish using the tool SurveyMonkey (https://es.surveymonkey.com/ (accessed on 15 February 2022)) and was completed by 211 participants (Appendix A). The cohort was categorized in early pregnancy (≤12 weeks of gestation; n = 32), pregnancy (>12 weeks of gestation; n = 119) and lactation women (n = 60).

### 2.2. Socioeconomic and Clinical Variables

The following data were self-reported by woman: maternal age (years), nationality (categorized by Spanish/non-Spanish), gestational age (weeks of gestation, when women was in the lactation period was answered regarding to the gestational age at delivery), assisted reproduction techniques (ART, yes/no), educational level (illiterate, primary degree, high school degree and university degree), employment status (working/unemployed/others, such as student), economic income (average monthly income per capita in the family core, considering the Spanish national average [21]) categorized as: no-income, <1000 €/month, 1001–2500 €/month, 2501–4000 €/month and >4000 €/month and civil status (single/married/others, such as engagement). 

In addition, they answered the following questions regarding to lifestyle habits: consumption of wine and beer (never/sporadic/daily), liquors (never/sporadic/daily), tobacco (never smoker/smoker >24 months or ex-smoker/smoker ≤24 months or smoker/passive smoker)- Tobacco index was calculated in smokers multiplying the number of cigarettes per day by the number of years smoked and divided by 20 [22]. Women also informed about ad-hoc perceived social support among family, partners, friends, or others, categorized on a numerical scale (the higher the score, the larger support perceived). 

Pregnant women answered if they had been diagnosed any of the following gestational complications: gestational diabetes, pregnancy-induced hypertension, preeclampsia and intrahepatic cholestasis. Women who reported gestational diabetes, pregnancy-induced hypertension or preeclampsia were categorized as severe obstetric complications. 

On the other hand, lactating women answered questions related to days of lactation and infant feeding practices, which were categorized, according to WHO criteria [23], as exclusive feeding on mother’s own milk or mixed breastfeeding (infant who had received predominantly formula along with breast milk or other milk, without complementary foods). Lactating women answered questions about the diagnosis of mastitis and nipple pain or cracking. 

For lactating women, a breastfeeding adherence score (BAS) was calculated. This instrument identifies women who may be at risk of breastfeeding cessation and was used previously in Spanish population with high reliability [8]. BAS is a four items instrument with binary and Likert responses (1–5). The items were related to educational level, previous experience on breastfeeding and expectations and feeling about breastfeeding. Total scores range from 0 to 12 with higher scores indicating higher risk of breastfeeding cessation. The cut-point of the BAS was established in ≥5 points, with a sensitivity of 80.0% and a specificity of 60.0% [24].

### 2.3. Instruments

**The psychological capital form (PsyCap)**. One of the most used instruments to evaluate the psychological capital developed by Luthans et al. [25]. In this study the version of 12 items was used. PsyCap is a second-order construct consisting of four subscales, each comprised of 3-items. The subscales include hope, self-efficacy, resilience and optimism. Some sample items for PsyCap are the following: “*I feel confident analyzing a long-term problem to find a solution*” (Self-efficacy); “*There are lots of ways around my problem*” (Hope); “*I always look on the bright side of things*” (Optimism); and “*I usually manage difficulties one way or another*” (Resilience). In this study, the response pattern followed a 6-point Likert scale ranging from 6 (totally agree) to 1 (totally disagree). Prof Wernsing reported an internal consistency range 0.85–0.93 for PsyCap in 12 countries [26]. The results presented in this article indicate that the PsyCap had an internal consistency of α = 0.90.

**The health survey form (SF-36).** The SF-36 is one of the most used and evaluated generic health-related quality of life. The SF-36 is composed for 35 items evaluating the follow scales physical functioning (10 items), physical role functioning (four items), bodily pain (two items), general health (five items), vitality/fatigue (four items), social functioning (two items), emotional functionality (three items) and general mental health (five items). Scores on the SF-36 scales are transformed to a 0–100 scale, with higher scores indicating better health status. In this article was used the Spanish version [27], which reported an overall internal consistency of 0.90 [28]. The results presented in this article indicate that the SF-36 has an internal consistency of α = 0.67.

**The Pittsburgh sleep quality index (PSQI).** The PSQI is a 19-item questionnaire designed to measure sleep quality and disturbance over the past month in clinical populations [29]. In this study, the version adapted to Spanish speakers was used [30,31]. The 19 items are grouped into seven components, including (1) sleep duration, (2) sleep disturbance, (3) sleep latency, (4) daytime dysfunction due to sleepiness, (5) sleep efficiency, (6) sleep quality and (7) sleep medication use (no evaluated in this study). Each of the sleep components yields a score ranging from 0 to 3, indicating the higher score the greatest dysfunction. The sleep component scores are summed to yield a total score ranging from 0 to 21 with the higher total score (referred to as global score) indicating worse sleep quality. In distinguishing good and poor sleepers, a global PSQI score >5 yields a sensitivity of 89.6% and a specificity of 86.5% [29].

**The pregnancy physical activity questionnaire (PPAQ).** The version validated for the Spanish population was used in this study [32]. The PPAQ is a semiquantitative questionnaire that reports the time spent in 32 activities categorized into five categories including household task (13 items), work activities (five items), sports/exercise practices (eight items), transportation (three items) and inactivity (three items) and total activity. It was considered the intensity and the duration in each activity to calculate the scores [33]. The higher score in the category the more physical activity. 

Respondents are asked to select the category that best approximates the amount of time spent in each activity per day or week during the current trimester. At the end of the PPAQ, an open-ended section allows the respondent to add activities not already listed. Sleeping is not included. In this work, quantitative corrections were used applying the weekly energy expenditure (metabolic equivalent of task (MET)/h or week) in the activity [34]. The MET is the objective measure of the ratio of which a person expends energy, relative to the mass of that person, while performing some specific physical activity compared to a reference, set by convention at 3.5 mL of O_2_/Kg/min. 

### 2.4. Statistical Analysis

Data analysis was performed using R software (version 4.1.1, R Core Team 2021, Vienna, Austria) within RStudio (Version 1.4.1717, RStudio, PBC, 2009–2021, Inc., Vienna, Austria) using the *rcompanion, dplyr, tidyverse, devtools, arsenal, compareGroups, rio* and *oddsratio* packages. 

The data were expressed as median and interquartile range [Q1; Q3] for quantitative variables. Sample size (n) and relative frequency (%) was used to described qualitative variables. The univariate analysis was performed by Kruskal–Wallis test with Dunnett post-hoc or Mann–Whitney U test depend on number of comparison groups. Fischer’s exact test was used to compared proportions. 

To evaluate the association between psychological capital, general health, sleep indexes and physical activity with pregnancy and lactation separately, linear generalized regression models were build considering to ≤12 weeks of gestation as the reference. The variables to be associated were introduced in the models if an error probability <0.10 was shown in univariate analysis. From each model coefficients (β) ± standard error (SE) was extracted. The *p*-Value was extracted from each association factor. In this study techniques to impute data were not used and a *p*-Value < 0.05 was considered significant.

## 3. Results

### 3.1. Cohort Contextualization 

No differences in maternal age were detected between groups. As expected, gestational age, answered as gestational age at delivery, was significantly higher in the lactating group. Overall, the groups were similar in sociodemographic variables, except for employment status (Table 1). The percentage of employment was higher in the group of pregnant women below 12 weeks of gestation compared to the rest of the groups, and unemployment was larger in the lactation compared to the other groups. No significant differences between groups were detected in any of the parameters related to lifestyle habits between groups. Women in the lactation period perceived lower social support compared the other groups (Table 1). 

Regarding breastfeeding habits in the lactation group, women had been 31.0 [22.5; 43.0] days of breastfeeding; 55.3% of the women had exclusive breastfeeding and 44.7% had mixed breastfeeding. The breastfeeding adherence score was 8.0 [6.0; 11.0] higher than the 5 points established as the risk of breastfeeding cessation.

In the group of pregnant women above 12 weeks of gestation, obesity was the most prevalent morbidity (17.3%) and gestational diabetes, the most common obstetrical complication (17.3%). In the lactation group nipple pain (80.0%) was the most common issue reported.

### 3.2. Psychological Capital and Health Survey

No statistically differences were found between groups in any of the categories evaluated, neither in the psychological capital nor in the general health survey form (Table 2).

### 3.3. Sleep Quality and Physical Activity

In relation to the sleep evaluation by PSQI, it was shown that women in the lactation period presented higher scores in sleep duration, efficiency and quality compared to pregnancy groups, indicating a worse sleep (Table 3). In relation to physical activity by PPAQ, it was found that women in the lactation group increase their household tasks with respect to the pregnancy groups. However, they had lower values in all the other activities than pregnancy groups, being significant activities that involve transportation. It should be noted that during lactation, the overall total physical activity was higher than during gestation, possibly due to the increase in household tasks, suggesting that during lactation, women lifestyle was restricted to the house environment (Table 3).

### 3.4. Association Factors during Pregnancy and Lactation

The ≤12 weeks of gestation group was considered as the reference to test the association between sleep and physical activity factors with women above 12 weeks of gestation and during lactation. The models were adjusted by employment and civil status, economical income and perceived social support. 

There were not association factors in sleep during pregnancy. However, a worse sleep duration was significantly associated with the lactation period (Table 4). The household tasks were significantly associated with the lactation period but not with the end of pregnancy. The decrease in activities that involve transportation were associated with both the end of gestation and the lactation period. In addition, the overall decrease in total activity was associated with the lactation period (Table 4).

## 4. Discussion

In this study, we evaluated several factors that may affect women well-being during maternity, such as mental health, sleep quality and physical activities and were compared how they affect women during pregnancy and lactation periods. Our data demonstrated that the women in the first month of lactation have a worse sleep quality and reduce their physical activities beyond those in the house environment, having a lower perceived social support, compared to women during pregnancy. These factors may negatively influence the well-being of women in the lactation period and could contribute to maternal postnatal anxiety and depression. 

Insomnia during pregnancy has been observed in some situations, associated with depressive symptoms [35]. A reduction in sleep quality is expected during pregnancy due to discomfort [36], despite the fact that along pregnancy there is an increase in melatonin secretion, with highest levels in the third trimester, decreasing abruptly after delivery [37]. Sleep hygiene may also be affected during lactation, although data are limited. On one hand, the reduction in melatonin after birth, suggests a possible decrease in sleep. 

Secondly, the care of the newborn and the potential new worries, such as breastfeeding, may also result in a poor sleep quality. In a study from Belgium, during lactation period women showed better sleep quality, but lower sleep efficiency during three first months after childbirth [38]. According to our data, the sleep duration, efficiency and quality were worse during lactation period, compared to women in pregnancy. Additionally, the worse sleep duration in women during lactation period compared to pregnancy was also demonstrated in our models.

A poor sleep can impoverish physical and mental health and even lead to depression [39]. Therefore, our results suggest the need of appropriate early counselling of women during postnatal period to improve their sleep patterns and reduce the risk of depression and other abnormal psychological outcomes, as previously suggested [40]. Midwives should plan interventions to improve maternal sleep quality, reduce fatigue levels and inform parents that breastfeeding is not a factor that reduces sleep quality or increases fatigue [14,41]. 

In addition, the women of our study did not perceive different in vitality/fatigue category of the SF-36. However, it is important related to the high risk of breastfeeding cessation of our cohort (BAS = 8.0 [6.0; 11.0]) and the low rate of exclusively breastfeeding found (55.3%). Additionally, our cohort of women during lactation demonstrated a reduction in social support, compared to the period of pregnancy, which may also increase the risk of poor psychological outcome. 

The importance of an appropriate body weight gain during pregnancy is well known to reduce the risk of preeclampsia and gestational diabetes [18]. In this sense, physical activity is promoted in pregnant women, and it has been demonstrated that even light physical activity protects against the postanal depression [42]. On the other hand, the sedentary behavior may promote anxiety symptoms immediately after childbirth [43]. During lactation, it has been demonstrated that moderate to vigorous physical activity does not negatively affect breast milk composition and volume [16]. 

In addition, the bone status was higher in women during lactation who were physically active, suggesting that daily physical activity might help to maintain good bone status [44]. According to our data, the overall score in physical activity for late pregnancy women was lower than early pregnancy women and lactation period. Although in our models, lactation women did not show significance, there was a trend in women in late stages of pregnancy. This could be due to the limitations of movement involved in the progression of pregnancy. 

However, we detected differences regarding the type of physical activity between pregnancy and lactation women, evidencing that during lactation period the woman is involved mainly in household tasks and reduced activity outside the domestic environment, such as doing sport or transportation tasks. A study in Vietnamese lactating women also showed that household task was associated up to 1.85 times with breastfeeding pattern [45]. According to our data, we estimated in 1.02 times-more household activities in lactation than during pregnancy. Several studies have demonstrated that household tasks decrease during pregnancy [46,47,48]. 

Lactation women shown lower transportation activity compared to pregnancy, similar trends were reported in other study [19]. Caring for the newborn may be a barrier that makes difficult for the mother to go away. Women who reported that childcare was a barrier were 1.7 times-more inactive and sedentary behaviors [49]. An essential recommendation would be daily walking of 40–50 min, strength and flexibility exercise, as light-to-moderate activities, which have been shown to improve the attitude and health-related quality of life [50]. 

This evaluated component may be related to inactivity, particularly important in women under C-section that is only recommended walking during first forty days. It is true that our models did not demonstrate significance, according to our univariate analysis, the metabolic expends energy in inactivity was lower in women during lactation than during pregnancy. Together with work activities and sports practices are categories with the lowest energy expenditure for lactating women. 

Interestingly, these categories involve social contact, which could lead a social disengagement during breastfeeding compared to pregnancy, being the social support one of the protective factors to support the breastfeeding adherence [51]. Although, the women of our study did not show different in social functioning category of the SF-36. There are studies that have assessed physical activity showing that it tends not to change until the sixth month postpartum, being children in the home, longer work hours and lack of childcare the predictors of becoming insufficiently active during or after pregnancy [49]. For this reason, the knowledge of physical activity categories to promote interventions previously, during and after pregnancy could be a primary strategy to improve health of the women. 

In the recommendations, it is necessary to remember to breastfeed prior to exercise, postponing breastfeeding to one hour after exercise, or expressing if required, in cases where infants are uncomfortable with feeding immediately after the mother exercises [52].

Furthermore, being the activity of lactating women restricted to the domestic environment may be detrimental. In an observational study in pregnant women was suggested an increased risk of depression to be related with high levels of household tasks, but this was attenuated being not significant after adjustment the models [53]. This aspect may be aggravated with a low perceive social support and unemployment in lactation women, as observed in the present study.

Interestingly to note, in the present study was that the SF-36 showed an improvement in all categories in relation to advanced gestation. Furthermore, hope, self-efficacy and resilience remained similar in all three groups, being optimism score a bit declined in women during lactation compared to pregnancy. The evidence shown that the maternal optimism declined from first to six months postpartum and positively scored correlated with adherence to healthy habits and [54] and negatively with exclusively breastfeeding cessation [8]. 

The optimism/pessimism dimension should be considered important aspects to be assessed during lactation. We suggest that studies addressing the role of sleep and physical activity in the peri- and postnatal period and its impact on women mental health remain necessary. Additionally, the point of view of positive psychology in the maternity has been poorly explored interesting to know the impact on maternal-neonatal health. It would be of value to have a psychological health sensor related to these spheres to improve physical-mental health in women during this vulnerable period of maternity. 

### Limitation and Future Directions

This was a cross-sectional study, and therefore, longitudinal information would be needed regarding to how the scores transited through all the stages in each woman. However, it is one of the few studies that compare two stages of woman’s life, such as pregnancy and lactation being a pilot study in areas, such as psychological capital, perceived health, sleep quality and physical activity. 

In the other hand, it may be necessary to determine the scores in a group of women who were not pregnant/lactation period. However, other study has been demonstrated that women without these conditions showed many different related to socioeconomics variables and healthy habits than pregnancy and lactation women [55]. Therefore, they may not be an appropriate group to explore due to their concerns and unlimited movement than during pregnancy and lactation.

Finally, it would be necessary more information from other areas related to ecological models of childcare [51,56] within bio-psycho-social approach [57,58,59]. Exploring the influence of twin pregnancies, neonatal sex, level of social support, or nutritional aspects would be key to gain a deeper understanding of the phenomenon of motherhood.

## 5. Conclusions

Overall, sleep duration deteriorated during lactation. In addition, we observed, in this period, an increase in household tasks and a decrease in other type of physical activities, particularly those outside the domestic environment. We suggest that the decline of physical activity during pregnancy, low perceived social support and social disconnection along with increased household task demonstrated during lactation can precipitate an unappropriated women’s mental health adjustment. Public health policies that improve maternity and paternity leave and greater employment flexibility can improve these conditions.

## Figures and Tables

**Table 1 ijerph-19-11199-t001:** Differences in sociodemographic characteristics between groups.

	≤12 Weeks of Gestation (n = 32)	>12 Weeks of Gestation (n = 119)	Lactation (n = 60)	*p*-Value
Maternal age (years)	32.0 [25.0; 35.0]	33.0 [29.0; 37.0]	33.0 [28.8; 36.0]	0.509
Gestational age (weeks) *	7.7 [5.9; 10.1] ^a^	27.0 [19.6; 34.0] ^b^	39.0 [38.0; 40.0] ^c^	<0.001
Origin				
Spanish	17 (58.6%)	65 (63.7%)	21 (45.7%)	0.119
Non-Spanish	12 (41.4%)	37 (36.3%)	25 (54.3%)	
Educational level				
Illiterate	1 (3.5%)	0 (0.0%)	0 (0.0%)	0.811
Primary degree	0 (0.0%)	3 (2.9%)	2 (4.4%)	
High School degree	12 (37.5%)	42 (35.3%)	15 (25.0%)	
University degree	16 (50.0%)	57 (47.9%)	29 (48.3%)	
Civil status				
Single	6 (20.7%)	5 (4.9%)	5 (10.9%)	0.066
Married	22 (68.8%)	95 (79.8%)	41 (68.3%)	
Others	1 (3.5%)	2 (2.0%)	0 (0.0%)	
Employment status				
Working	15 (46.9%)	43 (36.1%)	18 (30.0%)	0.005
Unemployed	5 (15.6%)	18 (15.1%)	29 (48.3%)	
Others	9 (28.1%)	30 (25.2%)	10 (16.7%)	
Economic status				
No-income	5 (17.2%)	8 (7.8%)	8 (17.4%)	0.082
<1000 €/month	4 (13.8%)	24 (23.5%)	13 (28.3%)	
1001–2500 €/month	14 (48.3%)	34 (33.3%)	19 (41.3%)	
2501–4000 €/month	4 (13.8%)	27 (26.5%)	6 (13.0%)	
>4000 €/month	2 (6.9%)	9 (8.8%)	0 (0.0%)	
Tobacco consumption				
Never	19 (65.5%)	60 (58.8%)	32 (69.6%)	0.774
Passive smoker	3 (10.3%)	18 (17.6%)	6 (13.0%)	
Ex-smoker	7 (24.1%)	19 (18.6%)	7 (15.2%)	
Smoker	0 (0.0%)	5 (4.9%)	1 (2.2%)	
Tobacco index	-	6.4 [6.4; 6.4]	7.5 [5.0; 7.5]	0.766
Wine and Beers intake				
Never	21 (72.4%)	87 (85.3%)	35 (76.1%)	0.127
Sporadic	7 (24.1%)	13 (12.7%)	11 (23.9%)	
Daily	1 (3.5%)	0 (0.0%)	0 (0.0%)	
Liquors intake				
Never	26 (89.7%)	97 (95.1%)	42 (91.3%)	0.408
Sporadic	2 (6.9%)	4 (3.9%)	3 (6.5%)	
Daily	1 (3.5%)	0 (0.0%)	0 (0.0%)	
Assisted reproduction techniques	5 (17.2%)	14 (13.7%)	5 (10.9%)	0.750
Perceived social support	4.0 [2.0; 6.0] ^a^	3.0 [1.0; 6.0] ^a^	1.0 [0.0; 3.0] ^b^	<0.001

Data show median and interquartile range [Q1; Q3] for quantitative variables and sample size and relative frequency (%) for qualitative variables. * At delivery was considered the gestational age in lactation group. The *p*-Values were extracted by Kruskal–Wallis test, Fischer’s exact test or Mann–Whitney U test. Different letters indicate significant differences by Dunnett post-hoc test. Sample size (n).

**Table 2 ijerph-19-11199-t002:** Differences in psychological capital and health indexes between groups.

	≤12 Weeks of Gestation (n = 13)	>12 Weeks of Gestation (n = 56)	Lactation (n = 21)	*p*-Value
**Psychological capital tool (PsyCap)**				
Hope	3.33 [3.33; 3.67]	3.67 [3.00; 4.00]	3.33 [3.00; 4.00]	0.931
Self-efficacy	3.00 [3.00; 3.33]	3.00 [3.00; 3.42]	3.00 [2.67; 3.67]	0.953
Resilience	3.00 [3.00; 3.00]	3.00 [2.67; 3.33]	3.00 [2.67; 3.67]	0.790
Optimism	3.00 [2.67; 3.00]	3.00 [2.58; 3.33]	2.67 [2.33; 3.33]	0.933
**Health survey form (SF-36)**				
Physical functioning	28.5 [24.0; 29.2]	26.5 [22.0; 29.0]	29.0 [23.5; 30.0]	0.257
Physical role functioning	6.50 [4.00; 8.00]	6.00 [4.75; 8.00]	8.00 [4.00; 8.00]	0.746
Bodily pain	6.15 [5.85; 8.20]	6.10 [5.20; 8.20]	7.10 [4.95; 7.20]	0.991
General health	11.7 [8.85; 14.3]	10.7 [8.40; 13.1]	11.0 [8.70; 12.4]	0.875
Vitality/fatigue	15.0 [12.8; 17.5]	13.5 [12.0; 17.0]	16.0 [12.5; 18.5]	0.460
Social functioning	5.00 [5.00; 5.25]	5.00 [4.00; 5.00]	5.00 [4.00; 5.00]	0.447
Emotional functionality	4.50 [3.00; 6.00]	5.50 [4.00; 6.00]	5.00 [3.00; 6.00]	0.555
General mental health	13.0 [12.5; 14.2]	12.5 [10.0; 16.2]	14.0 [12.5; 16.5]	0.524

Data show median and interquartile range [Q1; Q3]. The *p*-Values were extracted by Kruskal–Wallis test. Sample size (n).

**Table 3 ijerph-19-11199-t003:** Differences in sleep quality index and physical activity in between groups.

	≤12 Weeks of Gestation (n = 20)	>12 Weeks of Gestation (n = 71)	Lactation (n = 29)	*p*-Value
**Pittsburgh sleep quality index (PSQI)**				
Sleep duration	1.0 [0.0; 2.0] ^a^	1.0 [0.0; 2.0] ^a^	2.0 [1.5; 3.0] ^b^	<0.001
Sleep disturbance	10.0 [5.0; 12.0]	11.0 [9.0; 15.0]	9.0 [6.5; 12.5]	0.155
Sleep latency	2.0 [0.0; 3.0]	2.0 [1.0; 3.0]	1.0 [0.0; 3.0]	0.434
Daytime dysfunction	3.0 [2.0; 3.0]	2.0 [1.0; 3.0]	3.0 [1.5; 4.0]	0.348
Sleep efficiency	2.0 [0.0; 3.0] ^a,b^	0.0 [0.0; 2.0] ^b^	2.0 [1.0; 3.0] ^a^	0.021
Sleep quality	2.0 [2.0; 3.0] ^a,b^	2.0 [2.0; 3.0] ^b^	3.0 [2.0; 4.0] ^a^	0.041
Overall PSQI	21.0 [9.0; 24.0]	20.0 [14.0; 24.0]	19.0 [15.0; 25.0]	0.923
**Pregnancy physical activity questionnaire (PPAQ**; MET.h/week)				
Household task	296 [124; 1746] ^a^	267 [183; 927] ^a^	2312 [1238; 3344] ^b^	<0.001
Work activities	810 [0.0; 2827]	28.4 [0.0; 1295]	6.49 [0.0; 1260]	0.575
Sports practices	1.71 [0.10; 3.14]	1.14 [0.12; 3.50]	0.31 [0.05; 2.55]	0.291
Transportation	60.5 [5.38; 131] ^a^	31.7 [5.64; 73.6] ^a^	18.4 [0.0; 31.7] ^b^	0.001
Inactivity	241 [84.7; 778] ^a^	114 [27.0; 531] ^a^	27.0 [12.2; 209] ^b^	0.008
Total activity	2813 [1428; 3750] ^a,b^	2063 [775; 3105] ^b^	3452 [2252; 5218] ^a^	0.001

Data show median and interquartile range [Q1; Q3]. The *p*-Values were extracted by Kruskal–Wallis test. Different letters indicate significant differences by Dunnett post-hoc test. Sample size (n); Metabolic equivalent turnover (MET).

**Table 4 ijerph-19-11199-t004:** Association between sleep and physical activity with pregnancy and lactation.

	>12 Weeks of Gestation		Lactation	
	β ± SE	*p*-Value	β ± SE	*p*-Value
Sleep duration	−0.05 ± 0.34	0.873	1.13 ± 0.56	0.016
Sleep efficiency	−0.50 ± 0.37	0.176	0.29 ± 0.49	0.553
Sleep quality	−0.008 ± 0.27	0.977	0.22 ± 0.36	0.549
Household task	−33.46 ± 351.25	0.924	2147.3 ± 470.7	<0.001
Transportation	−69.66 ± 28.17	0.015	−78.55 ± 37.74	0.040
Inactivity	−111.13 ± 203.05	0.586	−99.31 ± 272.09	0.716
Total activity	−598.57 ± 513.46	0.247	1683.7 ± 688.05	0.016

Data show coefficients (β) ± standard error (SE). Models were adjusted by employment and civil status, economical income and perceived social support considering to ≤12 weeks of gestation women as reference group. The *p*-Value was extracted from the significance of each factor. The higher sleep index (PSQI) the worse sleep quality; the higher physical activity scores (PPAQ) the increase physical activities.

## Data Availability

The data presented in this study are available on request from the corresponding author. The availability of the data is restricted to investigators based in academic institutions.

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
