# Peer review of "Women during Lactation Reduce Their Physical Activity and Sleep Duration Compared to Pregnancy"

_ijerph, 2022, doi:10.3390/ijerph191811199_

Round 1
Reviewer 1 Report
One of my comments involves noting the high p values in some sections and maybe why this happened. This was an excellent idea to research because it demonstrates how we need to prepare women for breastfeeding--to me that was one of the most important findings. I always suspected it but no one had researched it. I wonder if maybe the quality of life survey you used wasn't too extensive? There are some shorter QOL surveys that could may be more relevant.
Author Response
One of my comments involves noting the high p-values in some sections and maybe why this happened. This was an excellent idea to research because it demonstrates how we need to prepare women for breastfeeding--to me that was one of the most important findings. I always suspected it, but no one had researched it.
Response: Thank you for your time reviewing our manuscript and your kind words. Undoubtedly, the perception of quality of life is a key factor that encompasses many areas of a woman's life and can condition aspects of health.
As for the high p-values in some variables, this could be due to the Kruskal-Wallis test. This test does not assume a normal distribution of the residuals and it is impossible to say, whether the rejection of the null hypothesis comes from the shift in locations or group dispersions. Therefore, the test is robust and less sensitive to outliers. However, only when the variable is stochastic and truly different between groups the p-value will be significant, otherwise the probability of error increases. We decided on this test because the sample size in some groups was limited, and we wanted to be more conservative in our conclusions.
I wonder if maybe the quality of life survey you used wasn't too extensive? There are some shorter QoL surveys that could maybe more relevant.
Response: That is right. Other questionnaires, such as the SF-12, are an adapted and shorter version of the one SF-36 used in this study. However, we wanted to explore the QoL extensively, as we thought that these could be interesting variables. The SF-12 assumes between 1-2 items to assess the QoL dimensions, we did not want our further data analysis to be limited to this fact and chose the full SF-36 where each dimension is explored by at least 4 items.
Reviewer 2 Report
I have evaluated the manuscript titled ‘Women during Lactation Reduce their Physical Activity and Sleep Duration compared to Pregnancy’. The researchers have focused on an important issue. The manuscript fulfills all the scientific standards necessary to be published. Results and Discussion are sufficient and well-organized. The title explains the content of the manuscript well and the abstract includes necessary and sufficient data. Further statistical analyses are not required. In fact, the analyses made by the researchers are sufficient. The language of the manuscript is good enough to understand and there are not any spelling or punctuation mistakes. The manuscript has sufficient quality and originality to be published in IJERPH.
Only authors should add an explanation about sample size calculation. It should write the effect size of the sample.
Also, you may attach your survey form to the supplement.
Respectfully yours,
Author Response
I have evaluated the manuscript titled ‘Women during Lactation Reduce their Physical Activity and Sleep Duration compared to Pregnancy’. The researchers have focused on an important issue. The manuscript fulfills all the scientific standards necessary to be published. Results and Discussion are sufficient and well-organized. The title explains the content of the manuscript well and the abstract includes necessary and sufficient data. Further statistical analyses are not required. In fact, the analyses made by the researchers are sufficient. The language of the manuscript is good enough to understand and there are not any spelling or punctuation mistakes. The manuscript has sufficient quality and originality to be published in IJERPH.
Response: Thank you for your time reviewing our manuscript and your kind words.
Only authors should add an explanation about sample size calculation. It should write the effect size of the sample. Also, you may attach your survey form to the supplement.
Response: The sample size calculation along with the statistical power was added to the text (lines 103-107). In addition, the complete form was attached as supplementary material.